# Interventional Few-Shot learning for ML Reproducibility Challenge 2020

## 1 Reproducibility Summary

### 2 Scope of Reproducibility

3 Main claim of the paper states that by using intervention ($P(Y|do(X))$ in few-shot learning (FSL) problem, we can
4 lower the bias coming from pre-trained models. Moreover the authors claim that they have an evidence of pre-trained
5 models being a confound variable in FSL tasks, meaning that relying too much on pre-trained features in training stage
6 based on support samples may lead to more errors in query set which is significantly different from support set. They
7 provide three different ways of implementing the intervention, based on backdoor adjustment for their Structural Causal
8 Model proposition. These are as follows:

9 • Feature-wise adjustment

10 • Class-wise adjustment

11 • Combined feature and class-wise adjustment

12 Report is focused on combined adjustment, as it gave best results in almost all of the cases mentioned in the paper.

13 Due to computing restraints (and some dataset unknowns) we decided to restrict our reproducibility to MTL (meta-
14 transfer learning) and SIB (Synthetic information bottleneck) settings on miniImagenet datasets.

### 15 Methodology

16 Authors provided the code, which can be found in `https://github.com/yue-zhongqi/ifsl` mini-ImageNet split
17 was taken from `https://github.com/hushell/sib_meta_learn` for SIB implementation and `https://github.`
18 `com/yaoyao-liu/meta-transfer-learning`, as stated by the authors in their repository. We changed some minor
19 elements of the pipeline to enable additional logging and hyperparameter optimisation. Trainings were generally done
20 on 1 RTX2080 GPU, no multi-gpu training was tested. The time needed to train one model varied from 5h for 1-shot
21 Resnet-based training of MTL to about 70h of training for IFSL 5-shot Resnet-based setting.

### 22 Results

### 23 0.0.1 SIB

24 For all the runs we used default hyperparameters that were provided in authors' code. One can see that for Resnet
25 architecture we achieve results slightly worse than the mean provided by the authors, however, excluding 1-shot
26 IFSL case, they fit in $3\sigma$ requirement. However, according to the paper, the most important thing is the performance
27 improvement. Authors report improvement in the range of 1.5-2 percentage points in accuracy on test set, however we
28 observe improvement in the range of 0.5-1 percentage points for all runs. It's worth to note that in all the cases the
29 performance is actually improved.

30 As compared to the paper, the results from the SIB algorithm on the test set are following:

| Setting used | Test acc 1-shot | Test acc 5-shot |
|---|---|---|
| Resnet10 Baseline - ours | 67.33 ± 0.59 | 79.01 ± 0.37 |
| Resnet10 IFSL - ours | 68.20 ± 0.56 | 79.93 ± 0.35 |
| Resnet10 Baseline - paper | 67.10 ± 0.56 | 78.88 ± 0.35 |
| Resnet10 IFSL - paper | 68.85 ± 0.56 | 80.32 ± 0.35 |

Table 1: Table with results for our run of SIB algorithm as compared to those showed in original paper. Values and confidence intervals are taken by averaging results over 100 subsamples (tasks) from the dataset.

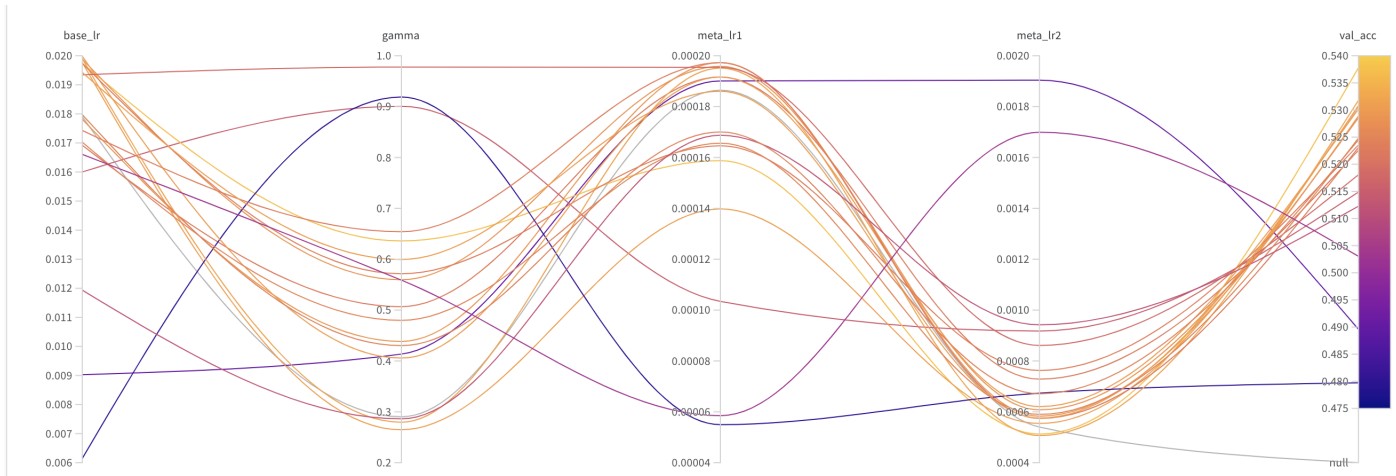

Figure 1: Result of hyperparameter tuning for MTL baseline. One can observe that as compared to default results shown in Table 1, optimisation either does not improve the accuracy in $3\sigma$ interval or is significantly worse.

### 0.0.2 MTL

As compared to the paper, the results from the MTL algorithm on the test set are following:

| Setting used | Test acc 1-shot | Test acc 5-shot |
|---|---|---|
| Resnet10 Baseline - ours | 53.01 ± 0.44 | 72.27 ± 0.36 |
| Resnet10 IFSL - ours | 60.16 ± 0.44 | 78.03 ± 0.34 |
| Resnet10 Baseline - paper | 58.49 ± 0.46 | 75.65 ± 0.35 |
| Resnet10 IFSL - paper | 61.17 ± 0.45 | 78.03 ± 0.33 |

Table 2: Table with results for our run of MTL algorithm as compared to those showed in original paper.

One can observe that the performance improvement in case of this setting is much higher (around 5 to 7 percentage points of accuracy), although baseline results are significantly lower than those provided by the authors.

We tried to perform hyperparameter optimisation, as compared to default parameters for results that were obtained on the validation set, we didn't observe significant positive difference, which can be observed on the Figure 1.

### What was easy

Running MTL and SIB algorithms was quite easy - for basic run provided instructions were sufficient. The only missing thing were sufficient configurable of paths - one had to look through all the project to change paths in a given file. Including additional logging and hyperparameter search with Weights and Biases framework didn't cause much trouble either.

### What was difficult

The reason why we focused only on MTL and SIB algorithms was due to reproducibility issues in other cases - errors such as missing *.npy files in case of MAML example, that were not to be found anywhere to download or configuration issues that happened when trying to run LEO algorithm.

There was a problem with consistency of miniImagenet dataset download - the main repository stated that one should download it using `https://github.com/hushell/sib_meta_learn`, whereas the subrepository with MAML code stated that on should use the `https://github.com/wyharveychen/CloserLookFewShot`, which required the download of whole ImageNet. Due to missing "novel.hdf5" file I couldn't reproduce results from MAML part of the paper. I described the issue in github repository `https://github.com/yue-zhongqi/ifsl/issues/4`.

The other thing to note is the length of training - it was possible to train the baselines in about 10-30 hours on single RTX2080 depending on basic architecture (Resnet or Wide-Resnet), however introducing intervention increased this time by a factor of 5, with less efficient GPU utilisation present.

### Communication with original authors

I tried to communicate with authors only by the repository with official code implementation, however I got no response. Link to the discussion can be found under the link:

https://github.com/yue-zhongqi/ifsl/issues/4

## Code and runs

Code: `https://github.com/freefeynman123/ifsl/tree/develop`

Runs: `https://wandb.ai/freefeynman123/mtl_ifsl_mini_imagenet?workspace=user-freefeynman123`

Hyperparameter tuning for baseline: `https://wandb.ai/freefeynman123/mtl_baseline_sweeps?workspace=user-freefeynman123`

