# OpenReview forum: "Interventional Few-Shot Learning"
_ML_Reproducibility_Challenge/2020 — Reject_

### Official Review · AnonReviewer1 · 2021-02-27
**Review, interesting but more can be said about hyperparameters and other experiment details**

**Rating:** 6
**Confidence:** 5

**Review:**

Summary: This is a replication of the NeurIPS paper "Interventional Few-Shot Learning" by Yue et al. (2020). This replicability-report focuses a lot on replicating four specific experiments from this paper: Resnet10 Baseline, Resnet10 IFSL using both the SIB and MTL algorithm for each (4 configurations). The report seems to corroborate the results for the SIB algorithm and obtain close but slightly different results for the MTL algorithm using different computational resources but the same code provided by Yue at al. (2020).

With respect to the pros, this replicability study is clear and easy to read. It is also very succinct. The report also contain all the main minimal expected elements for the Replicability Challenge, including the type of contact the author of the report had with the authors of the original paper. This report has information of what to expect when running the code provided by Yue et al. (2020).

On the other hand, a better detail of why the replication focused on only the four experimental cases could have improved the report. The original NeurIPS paper had a large evaluation protocol with various assumptions tested. Thus, this report provides a valuable but limited information about the original paper. Also, since the same code provided by the authors was used, it could have improve the replicability study if more information about the hyperparameter sweep, and may be other IFSL model assumptions, were added. This could have been done, for instance, by including a discussion of the parameters used to replicate the experiments. Finally, the report reads more like a summary of the study performed. In fact, there is really not distinct separation between the summary and the body of the report.

**Familiar With The Original Paper:**

I have read the original paper

**Reproducibility Summary:**

Report has summary

---

### Official Review · AnonReviewer2 · 2021-03-02
**Report consists solely of summary sheet**

**Rating:** 2
**Confidence:** 4

**Review:**

This short submission describes the author's attempts to reproduce Yue et al. (2020)'s "Interventional Few-Shot Learning" results. It focuses on one dataset (miniImageNet) and two methods (meta-transfer learning and simulated information bottleneck) that the original paper proposes can be augmented with the interventional strategy (IFSL). Using author-provided code, this submission reports that including IFSL does improve the model's performance, though not to the same extent reported in the original paper. A baseline discrepancy on the meta-transfer learning experiment was also observed.

Unfortunately, the submission includes very few details--it is essentially just the reproducibility summary--which limits the potential value of this work. The author obtained the best results using Yue et al.'s  hyperparameters; for all runs "the results were much worse" when these parameters were optimized by the author. However, the significance of this claim is tough to assess because no methodological details or quantitative results were reported in this submission. The only results are reported in Table 1 and 2 and these are unclear: over how many runs was the average taken? What is indicated after the ± (SD, SE, etc)?

The manuscript does not include any exploration or explanation of the results reported here, nor is the original method tested outside of the original framework. References to the broader literature (and the original paper!) are missing, and the writing could be more polished.

I regret that I cannot be more positive, but I do not believe the work is ready for publication.

**Familiar With The Original Paper:**

I have read the original paper

**Reproducibility Summary:**

Report has summary

---

### Official Review · AnonReviewer3 · 2021-03-10
**Interventional Few-Shot Learning**

**Rating:** 8
**Confidence:** 3

**Review:**


Thank you for this excellent work!

The authors clearly explain the Scope of Reproducibility.

The authors successfully conducted experiments to reproduce the paper. I would love to know if they use the same batch-size as the original paper?

I like that the authors conducted experiments on two different settings (MTL and SIB). In both experiments, I can see that the proposed approach improves baselines.

Issues regarding the report:
It would be better if authors can tell why there is a large gap between their numbers and original paper numbers for the MTL experiment.
It would be better if authors report experiment results with different hyperparameters.
Authors should try contacting the original paper author via their emails.

Overall, this is a nice work to reproduce the original paper and I recommend it for publication



**Familiar With The Original Paper:**

I have read the original paper

**Reproducibility Summary:**

Report has summary

---

### Decision · Program_Chairs · 2021-03-31

**Decision:**

Reject

**Comment:**

Overall reviews and/or the paper content not good enough for the AC to recommend to the journal.